# Sublethal thermal stress promotes migration and invasion of thyroid cancer cells

**Chi-Yu Kuo[1,2], Chung-Hsin Tsai[1,2], Jun Kui Wu[3], Shih-Ping Cheng** [ID][1,2,3,4]*

1 Department of Surgery, MacKay Memorial Hospital, Taipei, Taiwan, 2 Department of Medicine, School of Medicine, MacKay Medical College, New Taipei City, Taiwan, 3 Department of Medical Research, MacKay Memorial Hospital, Taipei, Taiwan, 4 Institute of Biomedical Sciences, MacKay Medical College, New Taipei City, Taiwan

* surg.mmh@gmail.com

## Abstract

### Objective

Radiofrequency ablation is a viable option in the treatment of benign thyroid nodules. Some reports suggest that thermal ablation may also be safe for the management of low-risk thyroid cancer. In this study, we applied transient heat treatment to thyroid cancer cells to mimic clinical scenarios in which insufficient ablation leads to incomplete eradication of thyroid cancer.

### Methods

Differentiated thyroid cancer cell lines B-CPAP, TPC-1, and FTC-133 were subjected to heat treatment at different temperatures for 10 min. Effects on cell growth, clonogenicity, wound healing assay, and Transwell invasion were determined.

### Results

Heat treatment at 45˚C or higher reduced cell growth, whereas viability of thyroid cancer cells was not changed after heat treatment at 37, 40, or 42˚C. Heat treatment at 40˚C increased the number of colony formations by 16% to 39%. Additionally, transient heat treatment at 40˚C resulted in a 1.75-fold to 2.56-fold higher migratory activity than treatment at 37˚C. Invasive capacity was increased after heat treatment, ranging from 115% to 126%. Expression of several epithelial-mesenchymal transition markers, including ZEB1, N-cadherin, and MMP2, was upregulated following heat treatment at 40˚C.

### Conclusion

We for the first time demonstrate that sublethal thermal stress may increase clonogenicity, migration, and invasion of thyroid cancer cells.

**Data Availability Statement:** All relevant data are within the manuscript and its Supporting Information file.

**Funding:** This work was supported by research grants from the National Science and Technology

Council of Taiwan (NSTC-111-2314-B-195-003
and NSTC-112-2314-B-195-009-MY3) and
MacKay Memorial Hospital (MMH-11215 and
MMH-11310). The funder had no role in study
design, data collection and analysis, decision to
publish, or preparation of the manuscript.

**Competing interests:** The authors have declared
that no competing interests exist.

## Introduction

Radiofrequency ablation (RFA) that induces thermal injury to target lesions has been shown to be a safe and effective alternative to surgery to treat benign, symptomatic thyroid nodules in selected patients [1, 2]. With a proper learning curve and precise technique, high volume reduction rates can be achieved [3]. Given the excellent results obtained from benign thyroid nodules, RFA is beginning to be adopted as an alternative option for the management of low-risk thyroid cancer. Although studies with mid-term follow-up yielded promising results with a low risk profile [4, 5], there was substantial heterogeneity with respect to the complete disappearance of thyroid cancer [6]. The long-term oncological safety of RFA treatment for thyroid cancer remains hypothetical.

Incomplete ablation of malignant tumors not only gives rise to residual lesions but may unexpectedly promote tumor progression. In hepatocellular carcinoma, multiple incomplete ablations were strongly associated with poor outcomes [7]. Some experimental evidence suggests that incomplete RFA may stimulate invasion, angiogenesis, and the development of cancer stem cells in malignant cells [8–10]. Although incomplete ablation was associated with adverse oncological effects in hepatocellular carcinoma, no study thus far investigated the effects of sublethal thermal stress on biological phenotypes in thyroid cancer cells. In this study, we applied transient heat treatment to thyroid cancer cells to mimic clinical scenarios in which an insufficient RFA leads to incomplete eradication of thyroid cancer. The aim of the present study is to examine whether sublethal thermal stress alters the malignant behavior of thyroid cancer cells.

## Methods

### Cell lines

In this study, we used three cell lines derived from differentiated thyroid cancers with different genetic backgrounds. Human papillary thyroid cancer cell line B-CPAP was obtained from the German Collection of Microorganisms and Cell Cultures (DSMZ), Braunschweig, Germany. Human papillary thyroid cancer cell line TPC-1 was obtained from Sigma-Aldrich, St. Louis, MO, USA. Human follicular thyroid cancer cell line FTC-133 was obtained from the European Collection of Authenticated Cell Cultures (ECACC), Salisbury, UK. The culture media used for B-CPAP and TPC-1 cells were Roswell Park Memorial Institute (RPMI) 1640 medium supplemented with 10% fetal bovine serum (FBS). FTC-133 cells were cultured in modified DMEM/F-12 medium with L-glutamine supplementation (Thermo Fisher Scientific, Waltham, MA, USA) and 10% FBS.

These cell lines have been validated to be of thyroid origin [11]. The genetic drivers include *BRAF* p.V600E for B-CPAP cells, *CCDC6-RET* fusion for TPC-1 cells, and *NF1* p.C167, *PTEN* p.R130, and *TP53* p.R273H for FTC-133 cells. We chose these cell lines because they were tumorigenic in immunocompromised mice based on our previous studies. This may facilitate subsequent animal studies. In our laboratory, cell lines were authenticated by short tandem repeat analysis, maintained at 37°C with 5% $CO_2$ at low passages, and tested periodically for mycoplasma contamination.

### Heat treatment

Thyroid cancer cells were suspended in complete media in microcentrifuge tubes. The microcentrifuge tubes were submerged in an isothermic water bath set to treatment temperatures for 10 min. The treatment duration of 10 min was chosen based on our previous study showing that the median RFA treatment time was 984 seconds with an interquartile range of 729 to

1256 seconds in the clinical setting [12]. We surmised that a portion of cancer cells would undergo sublethal thermal stress over a period of about 10 min. Cells following 10-min heat treatment were plated at 37˚C for subsequent experiments.

## Cell growth

Cell viability was determined using the Cell Counting Kit-8 (CCK-8) assay (Sigma-Aldrich) as we previously reported [13]. Following heat treatment, cells were counted and approximately 6,000 per well were seeded into 96-well culture plates. Cells were incubated at 37˚C in a humidified atmosphere for an additional 24 to 72 h. At indicated time points, 10 µl of the CCK-8 reagent was added into each well, and absorbance at 450 nm was measured using a multifunction microplate reader (Thermo Fisher Scientific) after incubation for 2 h at 37˚C. Five replicates were made for each measurement.

## Clonogenic assay

A colony-formation assay was carried out based on the method described previously [14]. Following heat treatment, thyroid cancer cells were seeded into 6-well plates at a density of 1,000 cells per well. At 8 days, cells were washed with phosphate buffered saline and fixed with a cold mixture of 3% acetic acid and 20% methanol for 30 min. After staining with 3% crystal violet dye at room temperature for 1 h, the plates were washed and dried. Colonies with more than 50 cells were visualized using a light microscope and manually counted.

## Wound healing assay

The wound healing assay is a simple and reproducible method of monitoring collective cell migration leading to wound healing [15]. Thyroid cancer cells were subjected to heat treatment, and a cell suspension at a concentration of about $5 \times 10^5$ cells/ml was prepared. The cell suspension (70 µl) was seeded into each well with the silicone culture insert of the µ-Dish (ibidi GmbH, Grafelfing, Germany). Cells were incubated at 37˚C with 5% $CO_2$ for 24 h to obtain a confluent cell layer. The culture insert was then gently removed with sterile forceps. At indicated time points, image acquisition was performed, and wound area was measured.

## Transwell invasion assay

Invasive capacity of thyroid cancer cells was examined using a Transwell chamber apparatus with Matrigel coating (Corning Inc., Corning, NY, USA), as previously described [16]. Briefly, the lower chamber was filled with culture media containing 10% FBS as a chemoattractant. Following heat treatment, B-CPAP ($7 \times 10^4$ cells per well), TPC-1 ($4 \times 10^4$ cells per well), or FTC-133 cells ($2.8 \times 10^4$ cells per well) were resuspended in serum-free media and seeded into Transwell inserts. After 24 h, cells invading through the membrane were fixed with methanol and stained with Diff-Quik (Sysmex, Kobe, Japan). Images were captured using a light microscope, and the number of invaded cells was counted manually.

## Immunoblotting

Following heat treatment, cells were re-plated and cultured for an additional 24 h. Total cellular protein was extracted from the cell pellet and electrophoresed in 12% polyacrylamide gel and transferred to a nitrocellulose membrane [17]. The membrane was blocked in 5% skim milk and incubated with the following primary antibody at 4˚C overnight: anti-ZEB1 (#70512; Cell Signaling Technology, Danvers, MA, USA), anti-Snail1 (PAB1924; Abnova, Taipei, Taiwan), anti-N-cadherin (#610921; BD Biosciences, Franklin Lakes, NJ, USA), anti-MMP2

(ab86607; Abcam, Cambridge, UK), and anti-GAPDH (MA5-15738; Thermo Fisher Scientific). Subsequently, the membrane was incubated with a secondary antibody conjugated with horseradish peroxidase at 37°C for 1 h. Chemiluminescent detection of proteins was captured with the Amersham ECL detection reagent (Cytiva, Marlborough, MA, USA).

## Statistical analysis

All data are expressed as mean ± standard deviation from a minimum of three independent experiments. GraphPad Prism version 8.3.0 (GraphPad Software, Boston, MA, USA) was used to evaluate the data. Statistically significant differences were calculated using one-way analysis of variance (ANOVA), and post hoc Dunnett's test was used for multiple comparisons between the values obtained at 37°C heat treatment and other conditions [18]. Differences with $P$-values of less than 0.05 were considered statistically significant.

## Results

To determine the effects of thermal stress on short-term cell growth of thyroid cancer cells, B-CPAP, TPC-1, and FTC-133 cells were treated with different temperatures ranging from 37°C to 50°C for 10 min. As shown in Fig 1, a 10-min heat treatment up to 42°C did not affect cell growth for the subsequent 72 h in all cell lines. A reduced cell growth rate was observed after a 10-min heat treatment at 45°C, while heat treatment at 47°C led to gradual decreases in cell viability. No viable cells could be detected at 48 h following a 10-min heat treatment at 50°C.

Next, we evaluated whether transient heat treatment affected subsequent clonogenicity in thyroid cancer cells. Heat treatment at 37°C did not alter the capacity of colony formation compared to cells without heat treatment. A 10-min heat treatment at 40°C increased the number of formed colonies by 16% in B-CPAP cells, 23% in TPC-1 cells, and 39% in FTC-133 cells, respectively, in comparison to heat treatment at 37°C (Fig 2). Clonogenicity was decreased after heat treatment at 42°C or 45°C. Heat treatment at 47°C or higher completely abolished colony formation in thyroid cancer cells.

The effects of heat treatment on cell migration were determined using the wound healing assay. There was no significant difference in cell migration between thyroid cancer cells subjected to heat treatment at 37°C and those without heat treatment. However, a 10-min heat treatment at 40°C resulted in a 1.75-fold higher migratory activity in B-CPAP cells, a 2.56-fold increase in TPC-1 cells, and a 2.56-fold increase in FTC-133 cells, respectively, compared to heat treatment at 37°C (Fig 3). Heat treatment at 42°C or 45°C suppressed migration in B-CPAP cells but not TPC-1 or FTC-133 cells.

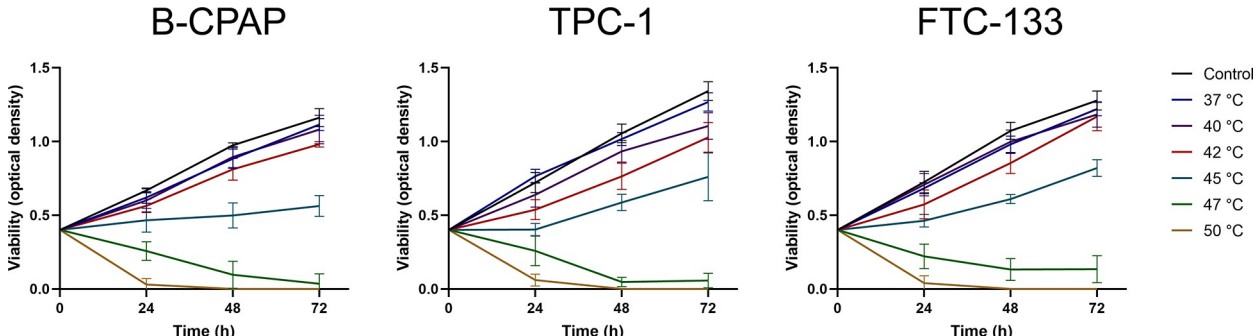

**Fig 1. Effects of heat treatment on cell viability in thyroid cancer cells.** Human thyroid cancer cell lines B-CPAP, TPC-1, and FTC-133 were subjected to heat treatment at different temperatures for 10 min, and cell viability was determined using the Cell Counting Kit-8 assay. The control group indicated parental cells without heat treatment. Each point represents the mean ± SD.

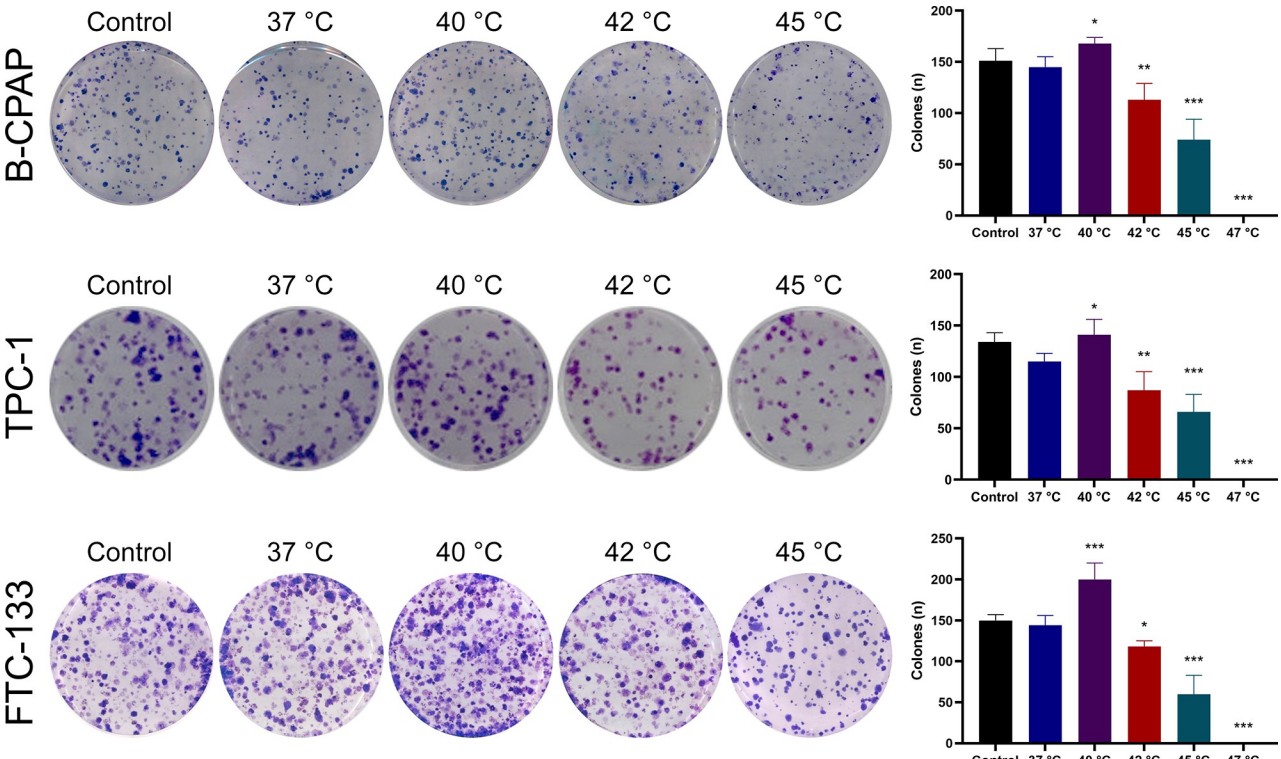

**Fig 2. Effects of heat treatment on clonogenicity in thyroid cancer cells.** Human thyroid cancer cell lines B-CPAP, TPC-1, and FTC-133 were subjected to heat treatment at different temperatures for 10 min, and colony formation was determined. The control group indicated parental cells without heat treatment. Each column represents the mean ± SD of the number of colonies. *$P<0.05$; **$P<0.01$; ***$P<0.001$, compared with the 37˚C heat treatment group (ANOVA post hoc Dunnett's test).

The Transwell assay was used to examine the effects of heat treatment on cell invasion. Invasive capacity after a 10-min heat treatment at 40˚C was 126% ± 8% of that after heat treatment at 37˚C in B-CPAP cells, 115% ± 11% in TPC-1 cells, and 125% ± 16% in FTC-133 cells, respectively (**Fig 4**). Invasive capacity decreased following heat treatment at 42˚C in B-CPAP and TPC-1 cells and heat treatment at 45˚C in FTC-133 cells. Taken together, transient sublethal heat treatment could significantly increase the invasiveness of thyroid cancer cells.

It seems that the most significant phenotypic effect of sublethal heat stress was associated with enhanced migratory activity. We assumed that sublethal heat stress may modulate the epithelial-mesenchymal transition (EMT) process in thyroid cancer cells. Therefore, we assessed the expression of several EMT markers in thyroid cancer cells at 24 h after heat treatment. As shown in **Fig 5**, the expression of EMT-activating transcription factor ZEB1, but not Snail1, was upregulated following a 10-min heat treatment at 40˚C compared to heat treatment at 37˚C. In concordance with ZEB1 upregulation, expression of mesenchymal markers N-cadherin and MMP2 was increased in thyroid cancer cells subjected to heat treatment at 40˚C. Altogether, these results suggest that sublethal heat stress promotes cancer cell migration and invasion, at least partly mediated by the modulation of EMT processes.

## Discussion

The incidence of thyroid cancer has consistently risen over recent decades [19]. The reason behind this upsurge is mainly overdiagnosis, in which subclinical thyroid cancers are increasingly identified through medical scrutiny. For these small lesions, active surveillance instead of

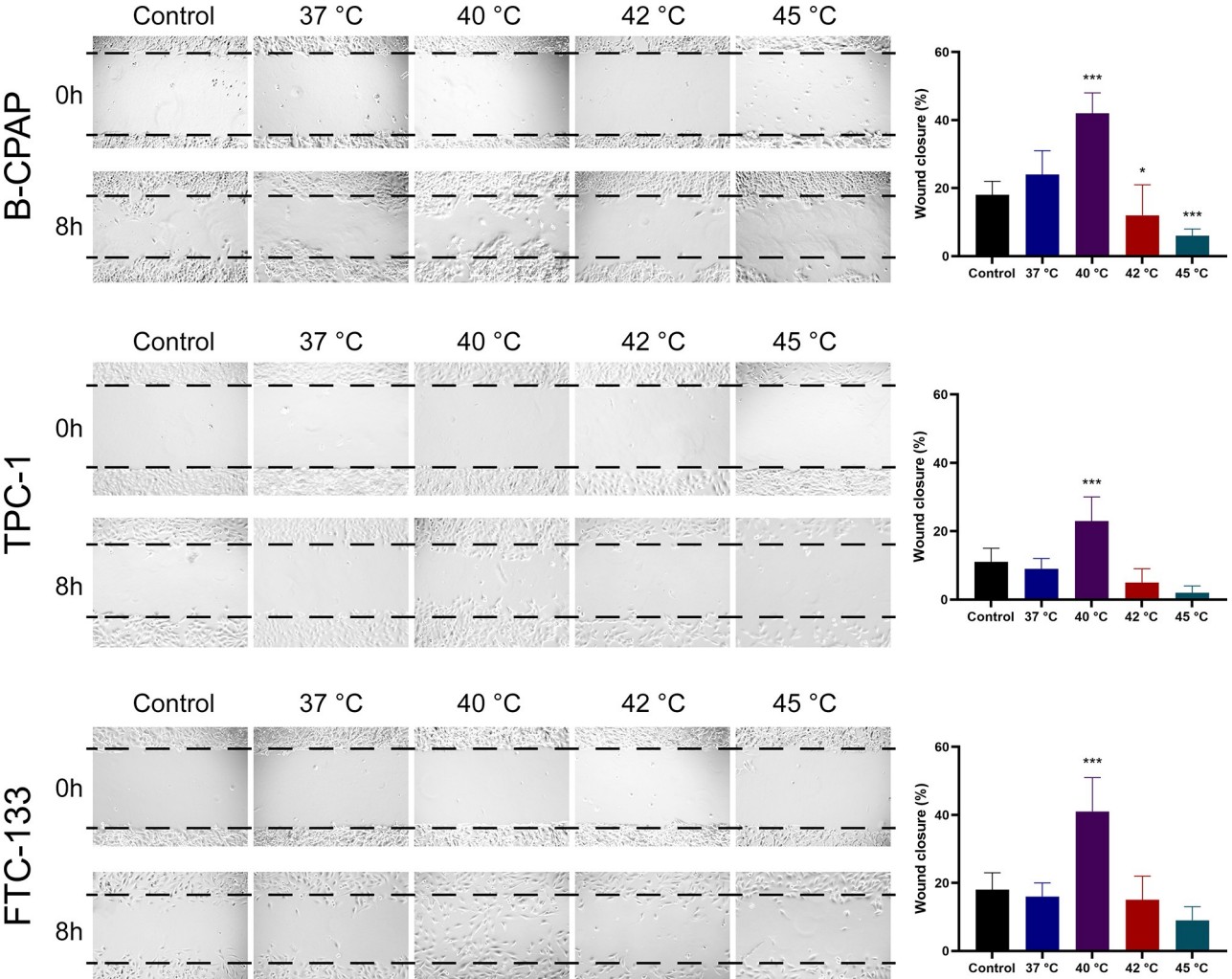

**Fig 3. Effects of heat treatment on cell migration in thyroid cancer cells.** Human thyroid cancer cell lines B-CPAP, TPC-1, and FTC-133 were subjected to heat treatment at different temperatures for 10 min, and cell migration was determined using the wound healing assay. The control group indicated parental cells without heat treatment. Each column represents the mean ± SD of the percentage of wound closures. *$P<0.05$; ***$P<0.001$, compared with the 37˚C heat treatment group (ANOVA post hoc Dunnett's test).

upfront surgery has been proposed as an appropriate management strategy for low-risk differentiated thyroid cancer [20]. By contrast, RFA is currently not considered to be a first-line treatment for primary thyroid cancer. The European Thyroid Association endorses minimally invasive treatments only in patients with radioiodine-refractory cervical recurrences who decline further surgery or are at surgical risk and in patients with unresectable oligometastatic or oligoprogressive distant metastases [21]. Nonetheless, thermal ablation has been proposed by some researchers as a potential treatment modality for low-risk papillary thyroid cancer because of its apparent safety and efficacy in terms of local control.

The thyroid gland is a relatively small organ compared to the liver, lungs, and kidneys and is surrounded by the trachea, esophagus, and cervical vessels. This anatomic constraint may lead to the incompleteness of the RFA of thyroid nodules. A previous study showed that locations close to the 'danger triangle' area or close to the carotid artery were potential factors associated with incomplete ablation of thyroid nodules [22]. Since current density decreases by the

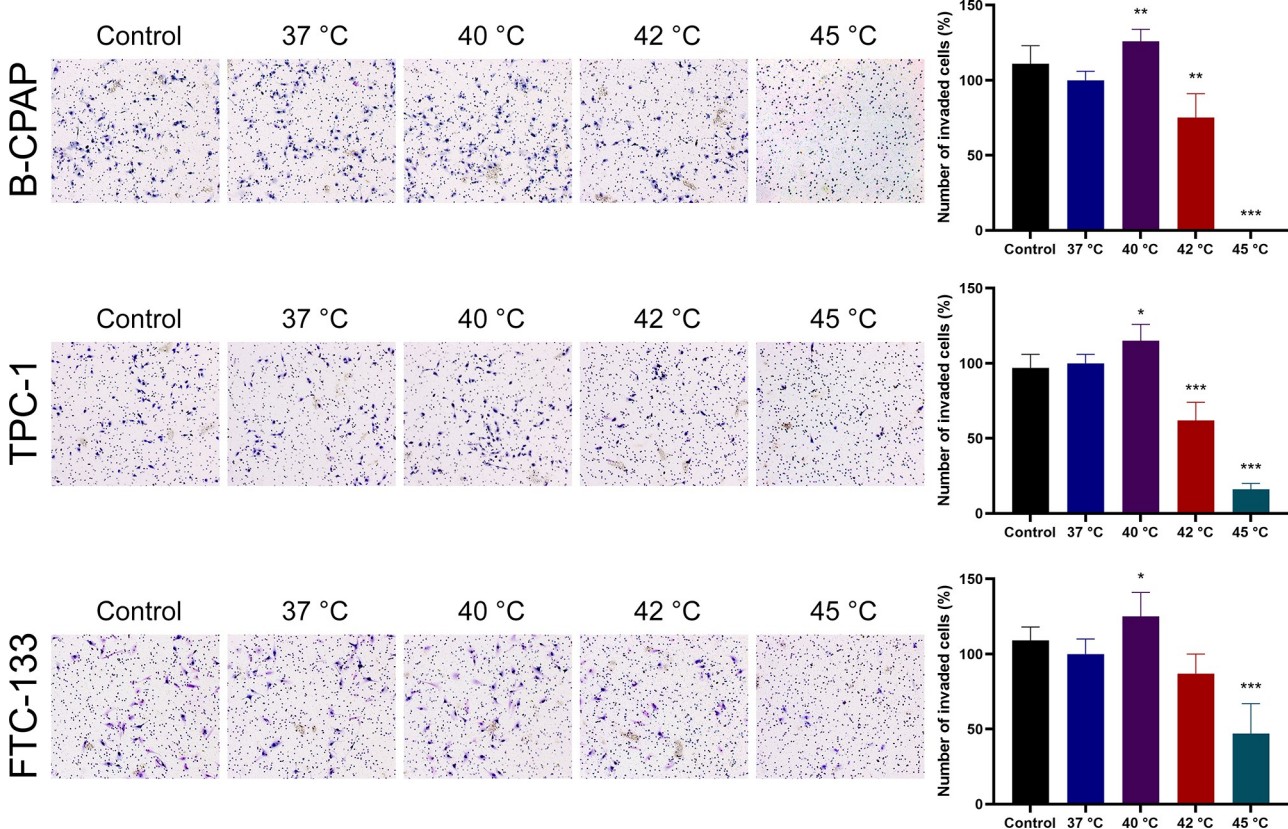

**Fig 4. Effects of heat treatment on cell invasion in thyroid cancer cells.** Human thyroid cancer cell lines B-CPAP, TPC-1, and FTC-133 were subjected to heat treatment at different temperatures for 10 min, and cell invasion was determined using the Transwell assay. The control group indicated parental cells without heat treatment. Each column represents the mean ± SD of the ratio of the number of invaded cells compared to that of the 37°C heat treatment group. *$P<0.05$; **$P<0.01$; ***$P<0.001$, compared with the 37°C heat treatment group (ANOVA post hoc Dunnett's test).

square of the distance to the ablation electrode, temperatures at the locations peripheral to the main lesion may be much lower than those adjacent to the electrode. Studies using ex vivo liver models have revealed that temperature distributions have wide variability [23]. Additionally, perfusion characteristics can affect the temperature distribution. In a study using tissue-mimicking phantom gel, the highest ablation volume was produced in hypovascular lungs whereas the lowest ablation volume was produced in the kidney as a highly perfused tissue [24]. Given the high vascularity of the thyroid glands, it would therefore be difficult to estimate safe margins of ablation.

In this context, a small series reported that viable residual thyroid cancer was present at histopathology in all of the 12 cases after percutaneous ablation [25]. In another study, two-thirds of malignant nodules after RFA were intraoperatively found to adhere to or invade the structures surrounding the thyroid [26]. It is also noteworthy that central lymph node metastasis is common in sporadic case reports of patients undergoing surgery for the residual tumor after RFA [27, 28]. Recently, radiologists from China demonstrated that incomplete response is the main risk factor for local tumor progression, while the incomplete response group showed 100% (13/13) tumor progression [29]. From this perspective, seemingly encouraging results may stem from the indolent nature of differentiated thyroid cancer. In line with this postulation, RFA showed no improvement in clinical symptoms or neck bulging among patients with anaplastic carcinoma [30].

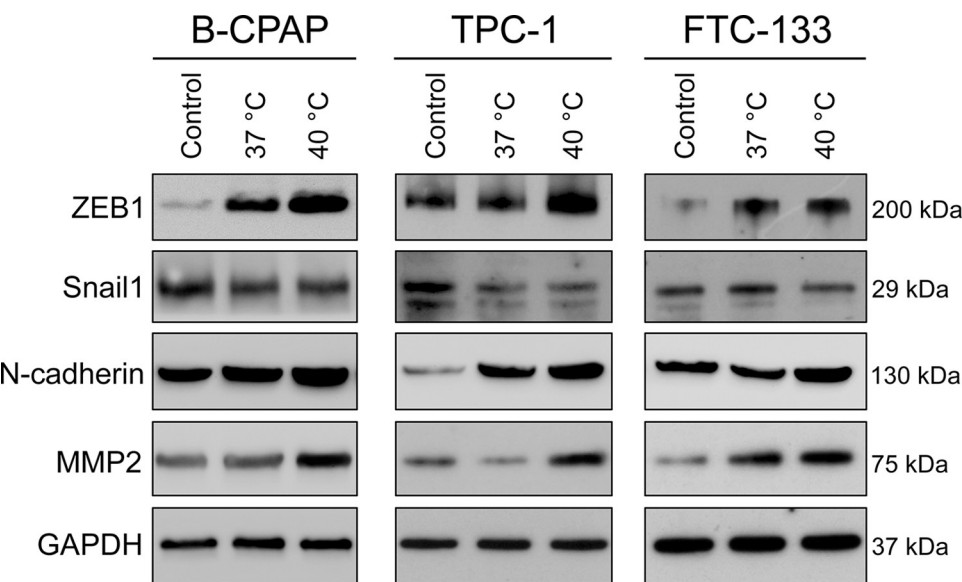

**Fig 5. Effects of heat treatment on expression of epithelial-mesenchymal transition markers in thyroid cancer cells.** Human thyroid cancer cell lines B-CPAP, TPC-1, and FTC-133 were subjected to heat treatment at 37°C or 40°C for 10 min. The control group indicated parental cells without heat treatment. Cell lysates were collected 24 h after heat treatment, and representative results from immunoblotting are shown.

To delve into the oncogenic effects of incomplete ablation in thyroid cancer, we designed this study and found that sublethal thermal stress at 40°C for 10 min significantly increased clonogenicity, migration, and invasion of thyroid cancer cells. Higher temperatures, even transiently, led to decreased cell viability and motility. Although heat treatment was reported to augment proliferation of heat-resistant sublines of hepatocellular carcinoma cells [8], we did not observe an immediate increase in cell growth after heat treatment in thyroid cancer cells. By contrast, the invasion-promoting phenomenon has previously been noted in liver cancer cells. Chen and colleagues demonstrated that transient heat stress induced an enhanced Warburg effect and glucose uptake in hepatocellular carcinoma cells, as well as an increase in cell invasion [31]. In another study, hepatoma cells exhibited increased invasiveness in response to conditioned media from tumor-associated endothelial cells exposed to 10-min heat treatment [9]. The results of these studies suggest an intriguing possibility that sublethal thermal stress from incomplete RFA treatment may stimulate the spread of neoplastic cells and may compromise disease outcomes in cancer patients. Although a recent retrospective study with 5-year follow-up indicated non-inferior rates of disease progression and recurrence-free survival in T1N0 patients undergoing RFA [32], it should be kept in mind that studies on differentiated thyroid cancer require a large sample size and long follow-up to achieve adequate statistical power [33].

The underlying mechanisms behind these tumor-promoting effects of sublethal thermal stress remain to be elucidated. Earlier studies have shown that hyperthermia at about 42°C is associated with irreversible inhibition of metabolism in several types of tumors with loss of biological activity, while at stimulatory temperatures between 42°C and body temperature, the metabolism and dissemination of tumor cells may be accelerated [34]. Biological effects of hyperthermia involve heat shock responses and alterations in cell membrane fluidity, thereby influencing the permeability and properties of membrane-bound proteins [35]. In particular, heat shock responses are accompanied by the induction of heat shock factors and chaperoning heat shock proteins that are implicated in multiple dimensions of tumorigenesis such as DNA

repair, metabolic regulation, and EMT changes [36]. We confirmed that heat treatment increased the expression of some EMT markers in the present study. However, the molecular processes linking sublethal thermal stress and heat shock responses to EMT activation in thyroid cancer need further in-depth investigation.

The strength of the present study is its consistent and reproducible results in all tested cell lines with a differentiated thyroid cancer origin and diverse genetic background. Regardless of the assorted oncogenic drivers, sublethal thermal stress had similar tumor-promoting effects in these thyroid cancer cells. Limitations include the monolayer cell culture and the lack of in vivo validation. Applying RFA in orthotopic or subcutaneous xenograft models would more closely resemble clinical RFA scenarios. In addition, detailed molecular mechanisms remain undetermined, and more mechanistic studies are needed. Despite these important caveats, the present study provides a new perspective on the implementation of thermal ablation in the treatment of low-risk thyroid cancer, where the completeness of tumor eradication and long-term oncological safety are still questionable.

In conclusion, we for the first time demonstrate that sublethal thermal stress may increase clonogenicity, migration, and invasion of thyroid cancer cells. It will be fascinating to assess the role of heat shock responses in tumor-promoting processes and the protective effects of chaperone-targeting interventions in the case of incomplete thermal treatment.

## Supporting information

**S1 Data. Minimal data set.**
(XLSX)

## Author Contributions

**Conceptualization:** Chi-Yu Kuo, Chung-Hsin Tsai, Shih-Ping Cheng.

**Data curation:** Chi-Yu Kuo, Jun Kui Wu.

**Formal analysis:** Chi-Yu Kuo, Shih-Ping Cheng.

**Methodology:** Chi-Yu Kuo, Jun Kui Wu.

**Writing – original draft:** Chi-Yu Kuo, Shih-Ping Cheng.

**Writing – review & editing:** Chung-Hsin Tsai, Jun Kui Wu.

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
