## [Decision Letter · Decision Letter 0]

19 Dec 2023

PONE-D-23-35946Sublethal thermal stress promotes migration and invasion of thyroid cancer cellsPLOS ONE

Dear Dr. Cheng,

Thank you for submitting your manuscript to PLOS ONE. After careful consideration, we feel that it has merit but does not fully meet PLOS ONE’s publication criteria as it currently stands. Therefore, we invite you to submit a revised version of the manuscript that addresses the points raised during the review process.

We look forward to receiving your revised manuscript.

Kind regards,

Yi-Hsien Hsieh, Ph.D.

Academic Editor

PLOS ONE

Journal Requirements:

This work was supported by research grants from the National Science and Technology Council of Taiwan (NSTC-111-2314-B-195-003 and NSTC-112-2314-B-195-009-MY3) and MacKay Memorial Hospital (MMH-11215 and MMH-11310). The funder had no role in study design, data collection and analysis, decision to publish, or preparation of the manuscript.

3. We note that your Data Availability Statement is currently as follows: All relevant data are within the paper.

Additional Editor Comments:

Reviewer-1

The research question is interesting and relevant to diverse thyroid cancer treatment trends. The authors tested the behavior of thyroid cancer cells in various temperatures and found the highest colony formation, migratory activity, and invasive capacity at 40 degrees. They suggest that sublethal thermal stress (which was 40 degrees) may increase clonogenicity, migration, and invasion of thyroid cancer cells. This information is vital for physicians choosing to apply RFA in thyroid cancer. However, several issues have to be clarified to make the deduction reasonable.

1. The choice of three cell lines with different genetic backgrounds is commendable, but the rationale for selecting these specific cell lines is not provided in the introduction. Including a brief justification for choosing these particular cell lines would strengthen the rationale.

2. The choice of a 10-minute heat treatment duration is explained based on the median RFA treatment time in clinical settings. However, based on the authors' information, the median treatment duration is around 15 minutes. Besides, the temperature range chosen for the heat treatment (up to 50 °C) should be discussed in the context of clinical relevance and safety. What is the potential range for a 40-degree in vivo? From the result, tumor cells are suppressed at 42 degrees or higher temperatures. Discussing how well these temperatures mimic the conditions encountered during RFA in thyroid cancer treatment would be beneficial.

Reviewer-2

Kuo and colleagues reported the first in vitro study in which thyroid cancer cells were subjected to sublethal heat stress for a transient time of 10 minutes. The authors found that migratory and invasive activity increased following heat treatment, accompanied by upregulation of epithelial-mesenchymal transition markers. The study has good scientific merit and a well-executed experimental design. The methods and statistical analysis appear appropriate. I have two minor comments on the study:

1. It may be helpful to validate the findings of cellular studies in animal experiments. This is a weakness of this study.

2. Recently, a response-to-ablation system was proposed (doi: 10.1007/s00330-023-10022-6). Incomplete response is the main risk factor for local tumor progression. This information may be added to the article.

Reviewers' comments:

Reviewer's Responses to Questions

**Comments to the Author**

1. Is the manuscript technically sound, and do the data support the conclusions?

Reviewer #1: Yes

Reviewer #2: Yes

2. Has the statistical analysis been performed appropriately and rigorously? 

Reviewer #1: Yes

Reviewer #2: Yes

3. Have the authors made all data underlying the findings in their manuscript fully available?

Reviewer #1: Yes

Reviewer #2: Yes

4. Is the manuscript presented in an intelligible fashion and written in standard English?

Reviewer #1: Yes

Reviewer #2: Yes

5. Review Comments to the Author

Reviewer #1: The research question is interesting and relevant to diverse thyroid cancer treatment trends. The authors tested the behavior of thyroid cancer cells in various temperatures and found the highest colony formation, migratory activity, and invasive capacity at 40 degrees. They suggest that sublethal thermal stress (which was 40 degrees) may increase clonogenicity, migration, and invasion of thyroid cancer cells. This information is vital for physicians choosing to apply RFA in thyroid cancer. However, several issues have to be clarified to make the deduction reasonable.

1. The choice of three cell lines with different genetic backgrounds is commendable, but the rationale for selecting these specific cell lines is not provided in the introduction. Including a brief justification for choosing these particular cell lines would strengthen the rationale.

2. The choice of a 10-minute heat treatment duration is explained based on the median RFA treatment time in clinical settings. However, based on the authors' information, the median treatment duration is around 15 minutes. Besides, the temperature range chosen for the heat treatment (up to 50 °C) should be discussed in the context of clinical relevance and safety. What is the potential range for a 40-degree in vivo? From the result, tumor cells are suppressed at 42 degrees or higher temperatures. Discussing how well these temperatures mimic the conditions encountered during RFA in thyroid cancer treatment would be beneficial.

Reviewer #2: Kuo and colleagues reported the first in vitro study in which thyroid cancer cells were subjected to sublethal heat stress for a transient time of 10 minutes. The authors found that migratory and invasive activity increased following heat treatment, accompanied by upregulation of epithelial-mesenchymal transition markers. The study has good scientific merit and a well-executed experimental design. The methods and statistical analysis appear appropriate. I have two minor comments on the study:

1. It may be helpful to validate the findings of cellular studies in animal experiments. This is a weakness of this study.

2. Recently, a response-to-ablation system was proposed (doi: 10.1007/s00330-023-10022-6). Incomplete response is the main risk factor for local tumor progression. This information may be added to the article.

6. PLOS authors have the option to publish the peer review history of their article (what does this mean?). If published, this will include your full peer review and any attached files.

Reviewer #1: **Yes: **Si-Yuan Wu

Reviewer #2: No

---

## [Author Response · Author response to Decision Letter 0]

6 Jan 2024

For Editor

Comment:

Response:

We have edited the manuscript to fulfill these style requirements.

Comment:

This work was supported by research grants from the National Science and Technology Council of Taiwan (NSTC-111-2314-B-195-003 and NSTC-112-2314-B-195-009-MY3) and MacKay Memorial Hospital (MMH-11215 and MMH-11310). The funder had no role in study design, data collection and analysis, decision to publish, or preparation of the manuscript.

Please provide an amended statement that declares all the funding or sources of support (whether external or internal to your organization) received during this study, as detailed online in our guide for authors at http://journals.plos.org/plosone/s/submit-now. Please also include the statement "There was no additional external funding received for this study." in your updated Funding Statement.

Response:

We moved the Funding Statement from the manuscript to the cover letter.

Comment:

3. We note that your Data Availability Statement is currently as follows: All relevant data are within the paper.

Please confirm at this time whether or not your submission contains all raw data required to replicate the results of your study. Authors must share the "minimal data set" for their submission. PLOS defines the minimal data set to consist of the data required to replicate all study findings reported in the article, as well as related metadata and methods (https://journals.plos.org/plosone/s/data-availability#loc-minimal-data-set-definition).

Response:

We uploaded our data set in EXCEL format as a supporting file.

Comment:

Response:

We revised the references to follow the journal's style.

For Reviewer #1:

The research question is interesting and relevant to diverse thyroid cancer treatment trends. The authors tested the behavior of thyroid cancer cells in various temperatures and found the highest colony formation, migratory activity, and invasive capacity at 40 degrees. They suggest that sublethal thermal stress (which was 40 degrees) may increase clonogenicity, migration, and invasion of thyroid cancer cells. This information is vital for physicians choosing to apply RFA in thyroid cancer. However, several issues have to be clarified to make the deduction reasonable.

Comment:

1. The choice of three cell lines with different genetic backgrounds is commendable, but the rationale for selecting these specific cell lines is not provided in the introduction. Including a brief justification for choosing these particular cell lines would strengthen the rationale.

Response:

We thank the reviewer for this comment. RFA as a treatment modality is confined to differentiated thyroid cancers, namely, papillary and follicular cancers/neoplasms. These cell lines were chosen to simulate clinical scenarios. Additionally, these cell lines are tumorigenic in immunocompromised mice based on our previous studies. This may facilitate subsequent animal studies.

Comment:

2. The choice of a 10-minute heat treatment duration is explained based on the median RFA treatment time in clinical settings. However, based on the authors' information, the median treatment duration is around 15 minutes. Besides, the temperature range chosen for the heat treatment (up to 50 °C) should be discussed in the context of clinical relevance and safety. What is the potential range for a 40-degree in vivo? From the result, tumor cells are suppressed at 42 degrees or higher temperatures. Discussing how well these temperatures mimic the conditions encountered during RFA in thyroid cancer treatment would be beneficial.

Response:

We thank the reviewer for this helpful suggestion. Previous studies showed that temperature distributions are affected by the distance and perfusion characteristics with wide variability. It is therefore difficult to estimate safe margins of ablation. We have expanded our discussion on this topic.

For Reviewer #2:

Kuo and colleagues reported the first in vitro study in which thyroid cancer cells were subjected to sublethal heat stress for a transient time of 10 minutes. The authors found that migratory and invasive activity increased following heat treatment, accompanied by upregulation of epithelial-mesenchymal transition markers. The study has good scientific merit and a well-executed experimental design. The methods and statistical analysis appear appropriate. I have two minor comments on the study:

Comment:

1. It may be helpful to validate the findings of cellular studies in animal experiments. This is a weakness of this study.

Response:

We thank the reviewer for pointing this out and have emphasized this weakness in our discussion.

Comment:

2. Recently, a response-to-ablation system was proposed (doi: 10.1007/s00330-023-10022-6). Incomplete response is the main risk factor for local tumor progression. This information may be added to the article.

Response:

We thank the reviewer for this suggestion and have included this article in the revised manuscript as reference #29.

---

## [Decision Letter · Decision Letter 1]

1 Feb 2024

Sublethal thermal stress promotes migration and invasion of thyroid cancer cells

PONE-D-23-35946R1

Dear Dr. Cheng,

We’re pleased to inform you that your manuscript has been judged scientifically suitable for publication and will be formally accepted for publication once it meets all outstanding technical requirements.

Kind regards,

Yi-Hsien Hsieh, Ph.D.

Academic Editor

PLOS ONE

Additional Editor Comments (optional):

Reviewers' comments:

Reviewer's Responses to Questions

**Comments to the Author**

1. If the authors have adequately addressed your comments raised in a previous round of review and you feel that this manuscript is now acceptable for publication, you may indicate that here to bypass the “Comments to the Author” section, enter your conflict of interest statement in the “Confidential to Editor” section, and submit your "Accept" recommendation.

Reviewer #1: All comments have been addressed

Reviewer #2: (No Response)

2. Is the manuscript technically sound, and do the data support the conclusions?

Reviewer #1: Yes

Reviewer #2: Yes

3. Has the statistical analysis been performed appropriately and rigorously? 

Reviewer #1: Yes

Reviewer #2: Yes

4. Have the authors made all data underlying the findings in their manuscript fully available?

Reviewer #1: Yes

Reviewer #2: Yes

5. Is the manuscript presented in an intelligible fashion and written in standard English?

Reviewer #1: Yes

Reviewer #2: Yes

6. Review Comments to the Author

Reviewer #1: The authors have addressed all the questions raised by the reviewers. Since the thermal spread in RFA treatment for thyroid nodules is variable (depending on peripheral perfusion), such findings could be an important consideration for physicians who want to apply RFA in thyroid cancer.

Reviewer #2: (No Response)

7. PLOS authors have the option to publish the peer review history of their article (what does this mean?). If published, this will include your full peer review and any attached files.

Reviewer #1: No

Reviewer #2: **Yes: **Shun Yu Chi

---

## [Editor Report · Acceptance letter]

13 Feb 2024

PONE-D-23-35946R1 

PLOS ONE

Dear Dr. Cheng, 

I'm pleased to inform you that your manuscript has been deemed suitable for publication in PLOS ONE. Congratulations! Your manuscript is now being handed over to our production team.

Kind regards, 

on behalf of

Dr Yi-Hsien Hsieh 

Academic Editor

PLOS ONE